# Molecular Docking Studies and In Vitro Activity of Pancreatic Lipase Inhibitors from Yak Milk Cheese

**DOI:** 10.3390/ijms26020756

**Published:** 2025-01-17

**Authors:** Peng Wang, Xuemei Song, Qi Liang

**Affiliations:** Functional Dairy Products Engineering Laboratory of Gansu Province, College of Food Science and Engineering, Gansu Agricultural University, Lanzhou 730070, China; wangpeng3335@163.com (P.W.); springwinter110@126.com (X.S.)

**Keywords:** yak milk cheese, peptide bioactivity, molecular docking, molecular dynamics simulation, pancreatic lipase inhibitory activity, pancreatic lipase inhibitory activity

## Abstract

Pancreatic lipase serves as a primary trigger for hyperlipidemia and is also a crucial target in the inhibition of hypercholesterolemia. By synthesizing anti-hypercholesterolemic drugs such as atorvastatin, which are used to treat hypercholesterolemia, there were some side effects associated with the long-term use of statins. Based on this idea, in the present study, we identified peptides that inhibited PL by virtual screening and in vitro activity assays. In addition, to delve into the underlying mechanisms, we undertook a dual investigative approach involving both molecular docking analyses and molecular dynamics simulations. The results showed that peptides RK7, KQ7, and TL9, all with molecular weights of <1000 Da and a high proportion of hydrophobic amino acids, inhibited PL well. Molecular docking and molecular dynamics showed that peptides RK7, KQ7, and TL9 bound to important amino acid residues of PL, such as Pro and Leu, through hydrogen bonding, hydrophobic interactions, salt bridges, and π-π stacking to occupy the substrate-binding site, which inhibited PL and identified them as potential PL inhibitors. In vitro tests showed that the IC_50_ of RK7 and KQ7 on PL were 0.690 mg/mL and 0.593 mg/mL, respectively, and the inhibitory effects of RK7 and KQ7 on PL were significantly enhanced after simulated gastrointestinal digestion. Our results suggested that peptides RK7 and KQ7 from yak milk cheese can be identified as a novel class of potential PL inhibitors.

## 1. Introduction

Hyperlipidemia (HLD) is a prevalent metabolic disorder regarded as a significant threat to human health, primarily characterized by heightened levels of triglycerides and cholesterol in the bloodstream [1]. Triglycerides and cholesterol represent two categories of lipids that circulate within the human bloodstream. Furthermore, excessive caloric intake results in elevated levels of triglycerides and cholesterol in the bloodstream, thereby amplifying the risk of heart disease and cardiovascular conditions. Pancreatic lipase (PL) is responsible for the hydrolysis of 50–70% of total dietary fat in the intestine [2]. It is produced by pancreatic acinar cells and specifically catalyzes the hydrolysis of triglycerides in the diet; this process mainly occurs in the small intestine, where PL, with the assistance of bile, breaks down triglycerides into monoglycerides and free fatty acids [3]. PL is the primary contributor to the onset of HLD and its associated complications. The targeted inhibition of metabolically essential enzyme activity, such as that of PL, has garnered considerable attention from researchers. This has stemmed from the fact that the inhibition of PL activity, as a potential strategy for treating HLD, primarily functions by restricting the bioavailability of dietary triglycerides and cholesterol. PL is an enzyme that facilitates the breakdown of fats within the small intestine. By inhibiting the activity of PL, we could consequently slow or limit the digestion and absorption of fats, thus reducing blood levels of triglycerides and cholesterol [4]. Currently, synthetic compounds, such as atorvastatin—a pharmaceutical agent utilized in the treatment of cardiovascular diseases—are employed as inhibitors of PL; however, the administration of these medications might be accompanied by side effects, including severe intestinal discomfort and muscular complications [5]. Consequently, it is imperative to investigate natural food-derived inhibitors of PL.

Yak has important socioeconomic significance in many plateau regions in the world, and yak milk is considered the staple food in plateau regions [6]. During the ripening period of yak cheese, with rennet and fermenting agents, the protein is hydrolyzed to produce bioactive polypeptides and amino acids [7]. A bioactive peptide is a unique protein fragment that is composed of 2–20 amino acids and has beneficial effects on human functions. Polypeptides have the characteristics of easy modification, low antigenicity, and rich physiological functions. They are a kind of functional factor with broad development prospects. Hydrophobic amino acids, such as Leu, Gln, Ile, and Pro, contained in polypeptides are closely related to the biological activity of polypeptides [8]. Studies have shown that a variety of bioactive peptides identified in yak cheese have many beneficial bioactive functions, such as anti-sugar [9], antibacterial [10], antioxidant, [11] and ACE activity inhibition [12].

Soybean peptides show lipid-lowering ability by inhibiting PL activity [13]. Bioactive peptides have emerged as promising agents in the prevention and management of hypolipidemia, given their efficacy and safety profile. Originating predominantly from intricate protein hydrolysates, their study has traditionally been hampered by lengthy research timelines and the complexities involved in isolating and analyzing individual methods. Virtual screening presents a transformative solution to these challenges by computationally simulating the binding affinity between target proteins and peptides.

Evidence from recent research indicates that this virtual approach has been effectively utilized in high-throughput screening to identify enzyme inhibitors, demonstrating its capability to forecast inhibitor–enzyme interactions with notable accuracy [14]. This computational technique not only accelerates the discovery process but also enhances the precision of identifying potential bioactive peptides, making it an indispensable tool in contemporary biochemical research [15,16,17]. In this experiment, PeptideDB and BIOPEP-UWM Database were used, and other bioinformatics platforms were combined to accurately and efficiently study the bioactivity of peptides. Discovery studio client v16.1.0 and ChemDraw professional 20.0 molecular docking software were used to rapidly and efficiently predict the bioactivity and mechanism of action between peptides and PL. GROMACS 2022.03 software [18] was used to conduct molecular dynamics simulations to study the structural stability, dynamic properties, and functions of the complex, and it jointly clarified the mechanism of action of bioactive peptides [19].

The central focus of this research was on three specific peptides: RK7 (RPKHPIK), KQ7 (KVLPVPQ), and TL9 (TPVVVVPPFL) derived from yak grandmother cheese. These peptides were meticulously isolated utilizing a chloroform–methanol extraction technique and subsequently refined through Sephadex G-25 dextran gel chromatography, as detailed in reference [8]. This methodological approach ensured the effective purification and separation of the peptides for further investigation. We analyzed the physicochemical properties and amino acid composition of peptides using bioinformatics tools such as NovoPro and Allpeptide. Virtual screening of PL inhibitory peptides in yak milk casein was conducted by combining molecular docking and molecular dynamics, and the molecular binding mechanisms of these peptides with PL were elucidated. In addition, the inhibitory activity of polypeptides on PL in vitro and the effect of simulated gastrointestinal digestion on PL inhibitory activity were also tested. This study aimed to improve our theoretical understanding of the mechanism of PL inhibition by peptides in yak milk cheese, promote the widespread utilization of yak milk cheese peptides, and provide technical support for creating safe alternatives to cholesterol-lowering drugs.

## 2. Results and Discussion

### 2.1. Analysis of Physicochemical Properties of Peptides

PL, an important enzyme for fat breakdown in the digestive system, has inhibitors with the potential to treat blood lipid-lowering, weight control, and metabolic syndrome improvement. As Table 1 shows, hydrophobic amino acids (e.g., Pro, Leu, Val) in RK7, KQ7, and TL9 might reduce the enzyme’s activity by competitively binding to its active site. Peptide biological activity is closely related to amino acid composition. RK7, KQ7, and TL9 have a relatively high proportion of hydrophobic amino acids (RK7: 42.86%; TL9: 88.91%). A high proportion might enhance peptide membrane structure affinity, increasing the inhibition effect. According to Martinez Villaluenga et al. [20,21,22], amino acid residue characteristics are critical for peptide inhibitory properties. The number of amino acids in RK7, KQ7, and TL9 (7–9) is consistent with that in camel milk protein hydrolysate PL-inhibitory peptides, as reported by Mudgil et al. [23], indicating structural similarities and possible similar mechanisms of action. Molecular weight and isoelectric point might also affect peptide biological activity. KQ7’s isoelectric point is 9.84, and TL9’s is 3.32, suggesting different stabilities and activities in different pH environments. A higher molecular weight (e.g., TL9: 968.19 Da) might mean a longer in vivo half-life, enhancing the inhibition effect [24]. In conclusion, the physicochemical properties of RK7, KQ7, and TL9 are closely related to their PL-inhibition ability. Parameters like amino acid composition, molecular weight, and isoelectric point collectively influence their biological activity, providing a theoretical basis for new PL inhibitor development.

### 2.2. Molecular Docking Analysis

As shown in Table 2, the synthesized peptides can bind up to 37 PL residues, indicating a higher likelihood of target protease binding for peptides (RK7, KQ7, and TL9). Figure 1 shows the optimal docking conditions for the interaction between peptides RK7, KQ7, TL9, and PL active sites, as well as the evaluation of the binding affinity of peptides RK7, KQ7, and TL9 to PL. According to the docking hypothesis, lower docking energy indicates stronger binding affinity between proteins and peptide molecules. We analyzed the binding mode of the optimal docking peptide RK7 within the PL active site. The results showed that the VinaScore recorded by peptides RK7, KQ7, and TL9 docked with PL were −7.601 kcal/mol, −7.268 kcal/mol, and −8.304 kcal/mol. RK7 formed a strong hydrogen bond with amino acid residue Asn320 at a distance of 3.44 Å; it formed hydrogen bonds with Gln220 at distances of 3.53 Å, respectively, and also formed van der Waals forces with Gln220; it formed hydrogen bonds with Gly321 at a distance of 3.96 Å; and it formed van der Waals forces with amino acid residues Pro194, Asp193, and Val222. Peptide RK7 formed salt bridges with Glu188, Asp193, and Asp196 of PL. Salt bridges were formed by electrostatic interactions between charged amino acid residues. The active site of PL contains a catalytic triad composed of Ser153 -His264 -Asp177, which has a positive promoting effect on its lipolytic activity. RK7 was located close to the PL active site and was observed to form hydrogen bonds with His264, thereby enhancing the specificity and affinity of peptides and enzymes [25,26]. KQ7 forms a strong hydrogen bond with PL amino acid residue Gln220 at a distance of 3.46 Å. Strong hydrogen bonds were formed with amino acid Glu188 at distances of 3.68 Å, 3.85 Å, and 3.94 Å, respectively. KQ7 forms hydrogen bonds with the PL amino acid residue Ser323. In the case of KQ7, hydrophobic interactions with substrate active sites Val322, Phe228, and Pro187 were observed. KQ7 bound to Pro194, Asp193, Val222, Ser195, Gln324, Cys286, Tyr327, Asn167, Thr221, Ser219, Leu189, Pro16, Gly185, Thr186, and Pro187 through van der Waals forces. These forces, although usually weak, collectively promoted the binding of peptides to PL. The significant electrostatic interaction between KQ7 and Glu188 residues further enhanced the binding mechanism. Meanwhile, TL9 formed hydrogen bonds with the amino acid residues Gln220, Ser323, and Val222 of PL at distances of 3.59 Å, 3.71 Å, and 3.41 Å, respectively. It was supplemented by electrostatic interaction with Arg191. Meanwhile, TL9 underwent hydrophobic interactions with amino acid residues Ile211, Ile210, Pro16, Val322, Pro194, Pro187, and Leu189. In addition, TL9 also formed van der Waals forces with Pro209, Ala15, Arg191, Glu188, Asp193, Ser195, Ser219, Gly185, Thr186, Arg23, Gln184, and Phe183. The binding characteristics of TL9 were synergistic interactions of hydrogen bonding, electrostatic, hydrophobic, and van der Waals interactions. Similarly, atorvastatin (a known inhibitor) and other enzyme inhibitors also exhibited similar inhibitory effects (atorvastatin formed hydrogen bonds with the amino acid residue Ser323 of PL at a distance of 3.87 Å; formed van der Waals forces while forming hydrogen bonds with the amino acid residue Glu188; formed two hydrogen bonds with the amino acid residue Pro194 and formed π-π interactions with Pro194 in hydrophobic interactions; and formed hydrogen bonds with Val222 at a distance of 3.40 Å. These interactions played a key role in stabilizing drug–enzyme complexes and enhancing binding affinity.). In this study, some hydrophobic interactions stabilized the peptides in the PL binding pocket. Similar binding interactions, including hydrophobic and polar interactions, involving amino acids Glu188, Gln220, Val322, Pro194, Thr221, Arg191, Ser323, Asp193, and Ser195, have not been reported; therefore, based on the above analysis, the experimental results revealed the potential mechanisms by which peptides RK7, KQ7, and TL9 inhibited PL activity, emphasizing the importance of hydrophobicity. Future research could focus on optimizing these peptides to enhance their inhibitory ability while combining more modern computational biology methods to conduct an in-depth analysis of their binding mechanisms [27].

### 2.3. Molecular Dynamics Analysis

#### 2.3.1. GROMACS Analysis of the RMSD, RMSF, Hydrogen Bonds, SASA, and Gibbs Free Energy Stability Analysis

To better understand the dynamic processes and stability of PL binding with peptides (RK7, KQ7 and TL9), we analyzed their interaction details during a 100 ns simulation trajectory. The investigation specifically utilized the RMSD [28] (Figure 2a) and the RMSF (Figure 2b) of α Carbon atoms to scrutinize the structural dynamics of PL in complex with peptides RK7, KQ7, and TL9. The average RMSD values for peptides RK7, KQ7, and TL9 stabilized at 0.25 nm, 0.24 nm, and 0.25 nm; and the three complexes containing peptides reached relative stability at about 40 ns after simulation. The peptides (RK7, KQ7, TL9) induced elevated RMSF values in specific regions encompassing residues 100–120, 185–190, 200–220, 45–267, 300–320, and 400–430. These structural differences suggest that the peptide may have displaced the active region, thereby preventing other substrate molecules from binding to PL. Ligand–protein interactions often instigate local structural modifications, which may increase the protein’s flexible domain, thereby affecting RMSF values [29].

Hydrogen bonding (Figure 2c) serves as a neutral indicator for maintaining the stability of protein–ligand complexes. During the molecular dynamics simulation, the number of hydrogen bonds in the PL–peptide complex fluctuated between 0 and 12. The specific numbers of hydrogen bonds were as follows: RK7 exhibited 1–6 hydrogen bonds (up to 12), KQ7 showed 1–5 hydrogen bonds (up to 8), and TL9 had 3–7 hydrogen bonds (up to 9). The dynamic changes in hydrogen bond quantities not only reflected the strength of binding affinity but also revealed the flexibility of interactions between proteins and peptides. In this experiment, although the number of hydrogen bonds fluctuated, the overall stability was maintained, possibly related to the participation of specific amino acid side chains and main chains in hydrogen bond formation. Studies have demonstrated that research on the stability of protein complexes emphasized the central role of hydrogen bonds in maintaining structural integrity, and an increase in the number of hydrogen bonds was generally positively correlated with enhanced stability [30,31,32]. Therefore, the support from the literature further validated the findings of this experiment.

SASA (Figure 2d) PL complex changes are directly related to the folding state and stability of the peptide [33]. Pancreatic lipase underwent similar SASA changes with peptides RK7, KQ7, and TL9, reaching relatively stable mean values of 238.51 nm^2^, 239.47 nm^2^, and 240.01 nm^2^, respectively. These values reflected the degree of exposure of the protein surface after binding with the peptides. It was noteworthy that the hydrophobic SASA exhibited higher fluctuations during the simulation, indicating that the exposure of non-polar residues was significantly influenced by the peptides. Experimental results showed that the peptides RK7 and KQ7 had higher SASA values compared to TL9. High SASA values usually indicate greater structural flexibility of the protein in solution. This suggested that the structure of PL might have been more dynamic and flexible when binding to RK7 and KQ7, making it easier to interact with these peptides. This flexibility could have been an important characteristic for PL in performing its function, facilitating its adaptation to different substrates and environmental conditions. From a biological perspective, changes in SASA not only affected the structural stability of PL but might also have been directly linked to its catalytic activity and substrate binding capacity. As a key enzyme in the digestive system, the normal functioning of PL is critical for the digestion and absorption of lipids. Peptides could regulate the enzyme’s active state by binding to PL and altering its SASA values, thereby affecting the lipid metabolism process. Moreover, the extent of exposure of non-polar residues (i.e., changes in hydrophobic SASA) might also have had significant impacts on the biological function of PL. The exposure of non-polar residues might have influenced the hydrophobic interactions between the enzyme and its substrate, thereby altering the enzyme’s catalytic efficiency and substrate specificity.

The three-dimensional Gibbs free energy landscapes for the complexes of PL-RK7, PL-KQ7, and PL-TL9 are shown in Figure 2e–g. Within the PL-RK7 complex, lower Gibbs free energy is characterized by Rotation Radius (Rg) values ranging from 2.62 to 2.69 nm, coupled with RMSD measures between 0.15 and 0.20 nm. Similarly, the PL-KQ7 complex exhibits reduced Gibbs free energy at Rg values from 2.63 to 2.67 nm, coinciding with RMSD dimensions spanning 0.16 to 0.20 nm. The PL-TL9 complex showcases minimal Gibbs free energy when Rg spans 2.62–2.66 nm, and RMSD fluctuates from 0.17 to 0.23 nm. Low Gibbs free energy typically correlates with the enhanced conformational stability of a complex. These energetically favorable states likely stem from efficient interactions encompassing hydrogen bonds, hydrophobic contacts, and electrostatic forces, combined with optimally folded and compact protein structures [34].

#### 2.3.2. MM/GBSA Calculations and Analyses

##### The MM/GBSA Approach for Free Energy Estimation in Protein-Ligand Complexes

The MM/GBSA stands as a computational technique pivotal for approximating free energy in the context of ligand studies within protein complexes [35,36]. In this experiment, by calculating the binding free energy (ΔGMMGBSA) between the peptides and the target protein, we were able to assess the interaction strengths and stabilities of different peptides (RK7, KQ7, TL9) with the target medium, as shown in Table 3. The results indicated that the ΔGMMGBSA values for RK7, KQ7, and TL9 were −55.66 Kcal/mol ± 2.20, −35.54 Kcal/mol ± 3.27, and −29.59 Kcal/mol ± 4.05, respectively. The binding free energies of RK7 and KQ7 were significantly lower than that of TL9, suggesting that RK7 and KQ7 provide stronger interactions when binding to the target protein, allowing them to play a more effective role in biological processes (such as signal transduction and metabolic regulation). The binding strength of peptides was generally directly related to their biological activity, indicating that RK7 and KQ7 might have greater potential in regulating the activity of certain biological pathways or target proteins. Although the data in Table 3 showed that the contributions of van der Waals interactions among the three were roughly consistent, the other interactions of RK7 and KQ7 (such as hydrogen bonds, ionic bonds, and hydrophobic interactions) might have been more advantageous for binding stability. Studies have shown that a decrease in binding free energy is associated with increased affinity [37]. This reinforced our hypothesis that RK7 and KQ7 peptides might have played an important role in biological functions.

#### 2.3.3. Dynamic Interaction Analysis from 0 to 100 ns via Molecular Simulations

In the entire 100 ns molecular dynamics simulation (Figure 3), the stability and compactness of peptide RK7, KQ7, and TL9 complexes with pancreatic lipase (PL) were mainly attributed to the combined effects of hydrophobic and electrostatic interactions between hydrophobic amino acids. The composite maintained its overall structure with minimal structural changes, highlighting its resistance in dynamic environments. The complex structure diagrams and interaction forces presented in the molecular dynamics simulations at 0 ns and 100 ns indicated a significant shift in hydrogen bonding and salt bridge interactions in the protein complex. These changes were usually related to changes in functional regulation or protein stability. The experiment also showed that RK7, KQ7, and TL9 bound stably to PL during the 100 ns molecular dynamics simulation process and once again verified the accuracy and correctness of our previous experimental results. Structural analysis was conducted using molecular dynamics simulations of the 100 ns process, and in-depth analysis was carried out on the interactions between RK7, KQ7, and TL9 and PL during the simulation period, revealing the sustained stability of the complex. The observed increase in salt bridge interactions suggested that potential adaptive changes might contribute to improving stability. These insights confirmed the predictive power of molecular simulations and provided new perspectives on the dynamics of protein–peptide interactions and the biological mechanisms involved.

### 2.4. Synthesis and Validation of PL Inhibitory Activity of Peptides RK7 and KQ7

#### 2.4.1. In Vitro Validation of RK7 and KQ7’s Inhibitory Efficacy

As illustrated in Figure 4, a clear correlation emerged between the inhibitory efficiency of both peptides and their mass concentration; an increase in mass concentration was accompanied by a corresponding enhancement in inhibition rates. These findings laid the groundwork for further exploration into the molecular mechanisms underlying PL inhibition by these peptides. The inhibition curve of atorvastatin on PL was y = −28.929x^2^ + 82.324x + 14.632, R^2^ = 0.9532, with inhibition rates ranging from 29.27–67.09% at concentrations of 0.2–1.0 mg/mL. The curve of the peptide RK7’s inhibition on pancreatic lipase was y = −5.6429x^2^ + 61.341x + 7.702, R^2^ = 0.9807, with its inhibition ranging from 19.74–62.32% at concentrations of 0.2–1.0 mg/mL. The curve of the peptide KQ7’s inhibition on pancreatic lipase was y = −9.0893x^2^ + 53.162x + 19.152, R^2^ = 0.9482, with its inhibition ranging from 31.45 to 61.31% at concentrations of 0.2–1.0 mg/mL. The inhibitory efficacy of peptides RK7 and KQ7 on pancreatic lipase (PL) markedly surpassed that of the control cohort across all specified concentrations, as evidenced by statistical significance (*p* < 0.05). As delineated by the inhibition kinetics, RK7 and KQ7 boasted IC_50_ values of 0.690 mg/mL and 0.593 mg/mL, respectively, compared to atorvastatin’s IC_50_ value at 0.495 mg/mL. Indepth in vitro validation experiments elucidated that both RK7 and KQ7 exhibited potent inhibition against PL, with a demonstrable enhancement in inhibitory potency correlating with increased peptide mass concentration. The in vitro activity validation corroborated these findings, further substantiating the peptides’ inhibitory impact on PL, which intensified as peptide mass concentration escalated. The distinct amino acid compositions and sequence arrangements of RK7 and KQ7 significantly influenced their biological activities, with hydrophobic amino acids playing a pivotal role in modulating PL inhibitory efficacy [38]. In comparison to established statins, RK7 and KQ7 distinguished themselves through commendable performance in molecular docking and kinetic simulations. Statins, including atorvastatin, exert their inhibitory effects on PL via a multifaceted interaction mechanism. The in vitro experimental outcomes revealed that RK7 surpassed KQ7 in terms of PL inhibitory activity. Additionally, the stabilization of the peptide–enzyme complex was significantly bolstered by electrostatic and van der Waals forces. These insights contribute novel perspectives for the forthcoming development of peptide-based PL inhibitors, offering a promising avenue for therapeutic innovation.

#### 2.4.2. Simulating the Effect of Gastrointestinal Digestion on PL Inhibition Activity

RK7 and KQ7 have effective inhibitory effects on pancreatic lipase (PL) after gastrointestinal digestion, which is of great significance in cholesterol management. Pancreatic lipase plays a key role in the process of fat digestion and absorption. Inhibition of pancreatic lipase can reduce the decomposition and absorption of fat, thus helping to control the level of blood lipids. It has potential value in preventing and treating hyperlipidemia-related diseases such as atherosclerosis.

During the 120-minute simulated gastrointestinal enzyme digestion process (Figure 5), the PL inhibition rates of RK7 and KQ7 gradually increased over time. This indicates that these two peptides can continue to exert inhibitory effects on PL during gastrointestinal digestion, which may be related to their stability in the gastrointestinal tract and their sustained interaction with PL. KQ7 showed a significant increase in PL inhibition at 90 min (*p* < 0.05), and the inhibition rate reached 61.31% at 120 min; the inhibitory ability of RK7 was significantly enhanced after digestion (44.34–62.35%). This indicates that at specific stages of the digestion process, changes in the structure of peptides or their interactions with PL occur that are favorable for inhibiting PL activity, such as exposure of active sites or optimization of peptide PL binding patterns. The significant resistance of RK7 to pepsin and trypsin may be due to its inherent stable conformation or the sustained presence of potent fragments that can maintain their affinity for PL binding despite degradation. This characteristic makes RK7 highly promising as a therapeutic entity or dietary supplement. It can maintain activity in the digestive environment of the gastrointestinal tract, counteract PL-mediated effects, and potentially regulate cholesterol and fat metabolism processes in the body by inhibiting PL activity. At 90 min, the inhibitory ability of both peptides significantly increased (*p* < 0.05), indicating that the enzymatic degradation of pepsin and trypsin triggers the release of potent peptide segments. These potent peptide segments may have special structures or amino acid compositions that give them stronger inhibitory ability against PL. For example, they may have higher hydrophobicity or positive charge density, allowing them to interact more effectively with the active sites or other critical regions of PL. These released components may have superior biological functions, thereby enhancing their inhibitory effect on PL. They may enhance their inhibitory effect through various means, such as forming more hydrogen bonds with PL, van der Waals forces, and other interactions to suppress PL activity, which is consistent with previous experimental results of molecular docking and molecular dynamics.

The resistance mechanism of RK7 to pepsin and trypsin may be unique to it. Although there are studies on peptide anti-PL in existing literature [39], the specific way in which the stable conformation or potent fragment of RK7 maintains binding affinity may be different from other peptides. Similarly, the inhibitory surge phenomenon of KQ7 at 90 min may also be unique to it, and further in-depth research is needed to investigate the special mechanisms behind it.

## 3. Materials and Methods

### 3.1. Materials and Reagents

Originating from mature yak milk cheese, the peptides RK7, KQ7, and TL9 were of notable interest in our research. For the purpose of in vitro validation assays, we se cured synthetic forms of peptides RK7 and KQ7 supplied by Sangon Biotech, a compa ny based in Shanghai, China. Enzymes and reagents critical to our investigative processes were sourced diligently. Specifically, porcine pancreatic lipase was provided by Jiangsu Jingmei Bio technology, located in Yancheng, China. Additionally, the inclusion of atorvastatin (purity: 99%), a significant pharmaceutical compound, augmented our experimental design. This medication was obtained from Qilu Pharmaceutical, a manufacturer with operations in Jinan, China.

### 3.2. Instruments and Equipment

The modeling and visualization tasks were executed using Discovery Studio Client version 16.1.0, courtesy of Accelys Inc., headquartered in San Diego, CA, USA. Molecular structure illustrations were handled via ChemDraw Professional 20.0, developed by CambridgeSoft Corporation based in Cambridge, MA, USA. Laboratory instruments necessary for experimental procedures were acquired domestically. Microplate reading accuracy was assured by the VersaMax model, and sample mixing efficiency was achieved using the VM 500S vortex mixer, both procured from Quan’an Experimental Instrument Company Limited located in Ningbo, China. Precise pH measurements were made possible through the PHS 3C pH meter, a product of Shanghai YiDian Scientific Instrument Co., Ltd., situated in Shanghai, China. Furthermore, the FR224CN analytical balance, ensuring high precision weighing, was supplied by Ohaus Instrument Co., Ltd. with its base in Shanghai, China. And finally, temperature control during experiments was maintained using the HWS26 thermostatic water bath obtained from Shanghai Yiheng Technology Co., Ltd., also based in Shanghai, China.

### 3.3. Experimental Methods

#### 3.3.1. Forecasting the Physicochemical Traits of Peptides RK7, KQ7, and TL9

To ascertain the peptides’ properties accurately, various computational platforms were employed. The Peptide Property Calculator was utilized to evaluate the isoelectric point (pI), the net charge, and the theoretical molecular weight of the peptides under study [40]. For assessing the instability indices of the seven peptides, the ExPASy Prot Param platform was selected, additionally, the hydrophobic characteristics and the percentage of hydrophobic amino acids within the three peptides were determined using the advanced capabilities of the PEPTIDE 2.0 PL platform [41].

This methodical approach ensured a comprehensive analysis of the peptides’ physicochemical properties, providing a solid foundation for subsequent investigations.

#### 3.3.2. Structure Adjustment and Optimization of RK7, KQ7, TL9 and PL

The graphical representations of RK7, KQ7, and TL9 peptides were crafted utilizing Chem Draw Professional 20.0, as depicted in Figure 6. These structures were then imported into Discovery Studio Client v16.1.0, where the Prepare Ligands tool under ‘Small Molecules’ was used to generate conformations. All compounds underwent energy minimization with the CHARMM force field, a preparatory step for molecular docking with receptors. In this study, the three peptides adopted an irregularly coiled conformation, indicating a high level of conformational adaptability in the unbound state. This flexibility facilitated interactions with targets like receptors, enzymes, or proteins, enabling structural adjustments. Differing from the typical alpha-helix or beta-sheet, these irregular coils were more versatile, potentially enhancing functional capabilities such as signaling, protein interaction, or receptor engagement. This adaptability is characteristic of biologically active peptides with specific functions, like antimicrobial peptides, which can engage various biomolecules.

Peptide RK7, with the sequence Arg-Pro-Lys-His-Pro-Ile-Lys, produced two enantiomers and one stereoisomer, while KQ7 (Lys-Val-Leu-Pro-Val-Pro-Gln) and TL9 (Thr-Pro-Val-Val-Val-Pro-Pro-Phe-Leu) each resulted in one enantiomer and one stereoisomer. This variation could be attributed to the distinct amino acid compositions affecting conformational dynamics. The presence of proline (Pro) and other residues with high rotational freedom, such as lysine (Lys), influenced the generation of enantiomers. These amino acids enabled swift local structural alterations. Each peptide yielded a stereoisomer, which could significantly impact biological activity, as different isomers might exhibit varying binding affinities or biological responses. Stereoisomer presence could dictate the peptides’ specific binding to targets, their binding efficacy, and their role in signaling or catalytic processes within biological systems.

References [42,43] should be consulted for further details on the role of specific amino acids and the implications of stereochemistry on peptide function.

The three-dimensional structure of pancreatic lipase (PL) is depicted in Figure 7. The crystal structure of PL, identified by PDB ID: 1ETH, was sourced from the RCSB Protein Data Bank https://www.rcsb.org (accessed on 30 March 2024). The obtained protein file was subsequently opened using the Discovery Studio Client v16.1.0 software. Within this software, we meticulously removed aqueous molecules and heteroatoms from the three-dimensional model. The protein conformation was then refined using the ‘Clean Protein’ feature available in the Macromolecules module. Following this, hydrogenation was performed on the protein, and all compounds underwent energy minimization under the CHARMM force field, which further refined the protein’s architecture.

#### 3.3.3. Molecular Docking

The peptides RK7, KQ7, and TL9 were molecularly docked to PL (PDB: 1ETH) in a sequential manner using the ‘Dock Ligands’ (LibDock) functionality within the ‘Receptor Ligand Interactions’ module of Discovery Studio Client v16.1.0 docking software. Upon analysis, no notable disparities were observed in the 3D distance constraints among the various peptide–enzyme complex models. Consequently, the selection of the best model was based on successful docking, leading to the generation of 3D and 2D structures of the peptide–enzyme complexes, LibDock Score docking scores, and the identification of docking active sites. This comprehensive approach facilitated the determination of the optimal orientation of the active sites between the peptide and PL [41]. Following the successful docking process and the creation of a 2D graph through the ‘View Interactions’ mode, Client v16.1.0 software enabled visual inspection of the amino acid residues within the peptide–enzyme complex along with the intermolecular interaction forces, including Van der Waals, hydrogen bonding, and hydrophobic interactions.

#### 3.3.4. GROMACS Molecular Dynamics Simulation

In this study, the molecular dynamics experiment was executed utilizing GROMACS 2022.3 software [43]. Preprocessing for peptides RK7, KQ7, and TL9 involved the application of a GAFF force field through AmberTools22. Gaussian 16W was employed for hydrogenation and the computation of RESP potentials specific to these peptides. The potential energy data were incorporated into the molecular dynamics system’s topology file.

Simulation parameters were set at a constant temperature of 300 K and an atmospheric pressure of 1 bar. Amber99sbildn served as the chosen force field, with water molecules represented by the TIP3P water model. To maintain neutrality, an appropriate quantity of Na^+^ ions was introduced into the simulation system. Energy minimization was achieved using the steepest descent method, preceded by 100,000 equilibration steps under both constant temperature and volume (NVT) conditions, as well as constant temperature and pressure (NPT) conditions. Each equilibration phase lasted 100 ns with a coupling constant of 0.1 ns. The final step involved a free molecular dynamics simulation consisting of 5,000,000 steps, with each step lasting 2 ns, culminating in a total simulation duration of 100 ns.

Upon completion of the simulation, trajectory analysis was conducted using GROMACS 2022.3. Each peptide and protein in molecular dynamics is a complex system, and this complex system is repeated three times in a 100 ns molecular dynamics system. This analysis aimed to compute various parameters, including the root mean square deviation (RMSD), root mean square fluctuation (RMSF), hydrogen bonds (HBond), solvent-accessible surface area (SASA), and the binding free energy (MM/GBSA) for each amino acid’s motion trajectory. This comprehensive investigation aimed to assess the binding interactions between the enzyme and the peptides, thereby evaluating the interaction forces at play.

#### 3.3.5. PL Inhibition Experiments with Peptides RK7 and KQ7

Due to its highly hydrophobic properties, as evidenced by the computer simulations conducted, the peptide TL9 proved to be resistant to both synthetic purification and production. In the course of our investigation, two peptides, namely RK7 and KQ7, underwent solid-phase synthesis. Following this, the purity and molecular weight of the synthesized peptides were scrupulously examined using Liquid Chromatography–Mass Spectrometry (LC-MS). This entire experimental process was carried out by Sangon Biotech (Shanghai, China) Co., Ltd.

This experiment, based on the method of Glisan et al. [44] and with slight modifications to the experimental method, determined the effects of peptides RK7 and KQ7 on porcine pancreatic lipase (PL). Firstly, dissolve p-nitrophenylbutyrate (5 mM) in a sodium acetate solution (5 mM, containing 1% Triton X-100). Dissolve porcine pancreatic lipase (PL) (1 mg/mL) in 100 mM Tris HCl buffer. Incubate with different concentrations of porcine pancreatic lipase (PL) and peptides in 96-well plates (37 °C) for 30 min. Then, mix with 5 mM p-nitrophenylbutyrate solution to initiate the enzyme reaction. The labeled pig pancreatic lipase antibody binds to form an antibody–antigen enzyme-labeled antibody complex. The substrate is converted to blue by enzyme catalysis and ultimately to yellow by acid action. The intensity of color is positively correlated with the presence of porcine pancreatic lipase (PL) in the sample. The enzyme-linked immunosorbent assay (ELISA) was used to measure the absorbance (OD value) at a wavelength of 450 nm, and the activity concentration of porcine pancreatic lipase (PL) in the sample was calculated using a standard curve.(1)$$\mathrm{PL}\;\mathrm{inhibition}\;\mathrm{rate}/\%=\left(1-\frac{A-B}{C-D}\right)\;\times \;100$$
where *A* is the absorbance value of the sample; *B* is the absorbance value of the sample blank; *C* is the absorbance value of the control group; and *D* is the absorbance value of the control blank group.

#### 3.3.6. Simulating the Impact of Gastrointestinal Digestion on PL Inhibitory Activity

Peptides RK7 and KQ7 were dissolved in pepsin solution (enzyme–substrate 1:50 *w*/*w*). A concentrated HCl solution of 1 mol/L was added to the peptide enzyme mixture to adjust the pH to 2, which was incubated in a water bath at 37 °C for 90 min. Subsequently, NaOH solution with a concentration of 1 mol/L was added to adjust the pH to 7.5, followed by the addition of a mixture solution of trypsin and peptides (enzyme–substrate 1:50 *w*/*w*). The mixture was then incubated in a water bath at 37 °C for 180 min. The incubated solution was then boiled in a 95 °C water bath for 10 min to neutralize the pH. Samples were collected every 30 min and stored in a freezer at −20 °C [45].

#### 3.3.7. Data Processing

All measured data in the experiment were determined independently and measured in triplicate. All data were processed using SPSS Statistics 26.0. Data plots were generated using Origin 8.5 and GraphPad Prism 8.0.2.

## 4. Conclusions

This study indicates that the peptides RK7, KQ7, and TL9 produced through degradation in yak milk cheese are significant in inhibiting PL activity. This study confirmed its inhibitory activity on PL through different methods. In addition, we have demonstrated for the first time that yak cheese-degrading peptides have an effective inhibitory effect on PL. The inhibition rates of peptides RK7 and KQ7 on PL were 19.74–62.32% and 31.45–61.31%, and the IC_50_ values of inhibition were 0.690 mg/mL and 0.593 mg/mL. Meanwhile, molecular docking and molecular dynamics indicate that key amino acid residues, hydrogen bonds, and hydrophobic interactions are key factors in the binding process between peptides and PL. These results demonstrate the potential natural lipase inhibitors of peptides RK7 and KQ7, which can help control cholesterol as a risk factor for cardiovascular disease and provide theoretical support for future research on pancreatic lipase inhibitory peptides.

Although our future research focus has identified the key roles of key amino acid residues, hydrogen bonds, and hydrophobic interactions in the peptide–PL binding process, we can further investigate how these factors synergistically affect the inhibitory effect. The study of the specific impact mechanism of peptide binding to PL on lipid metabolism pathways at the cellular or animal level is not limited to cholesterol control but also considers its role in the entire lipid metabolism network. Animal model experiments should be conducted to evaluate the actual effectiveness of these peptides in preventing and treating cardiovascular diseases, including their long-term effects on blood lipid levels, vascular function, and other aspects. At the same time, we will further investigate how to apply these peptides to the development of functional foods, such as the amount and form of addition in low-fat dairy products or other healthy foods, as well as their impact on product taste and stability.

## Figures and Tables

**Figure 1 ijms-26-00756-f001:**
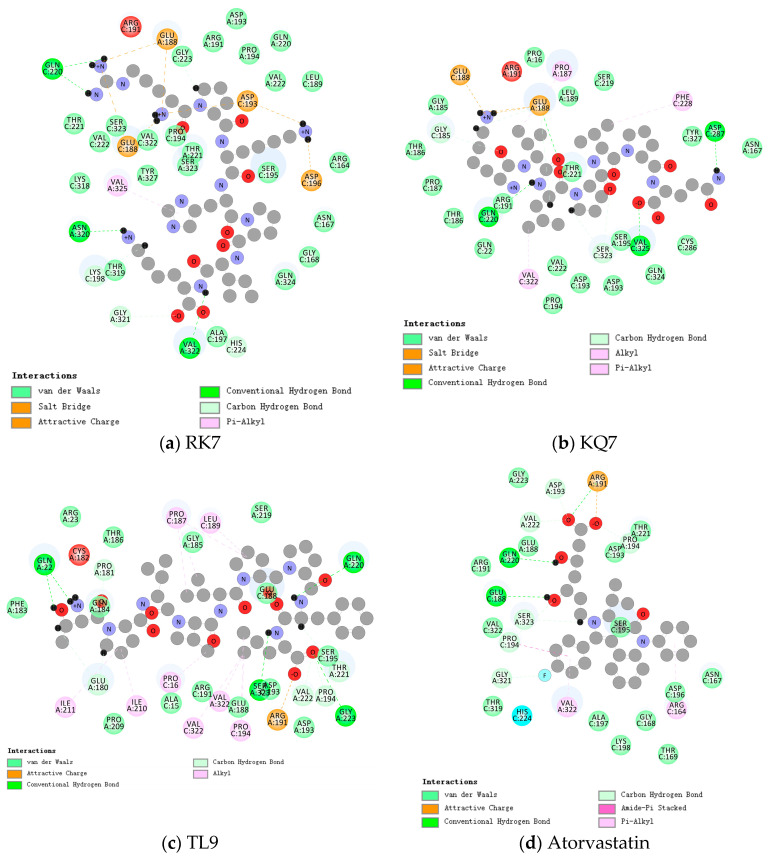
Close-up 2D view of the active site of PL bound to (**a**) RK7, (**b**) KQ7, (**c**) TL9, and (**d**) atorvastatin. Key residues of the peptide interacting with atorvastatin and PL binding are shown as bars and marked in green. Green dashed lines represent hydrogen bonds. Pink dashed lines represent hydrophobic interactions.

**Figure 2 ijms-26-00756-f002:**
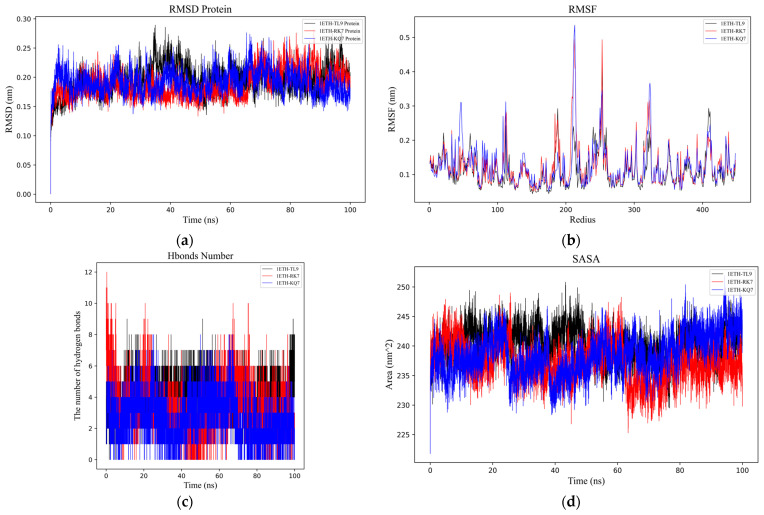
Analysis of molecular dynamics simulations for the peptide–PL complexes. (**a**) Depiction of the root mean square deviation (RMSD) trajectories for KQ7 (represented by the blue line) and RK7 (the red line) alongside TL9 (illustrated with the black line). (**b**) Display of the root mean square fluctuation (RMSF) profiles for KQ7 (denoted by the blue line), RK7 (signified with the red line), and TL9 (portrayed with the black line). (**c**) Presentation of hydrogen bond dynamics involving KQ7 (identified by the blue line), RK7 (marked with the red line), and TL9 (emphasized by the black line). (**d**) Illustration of solvent-accessible surface area (SASA) fluctuations for KQ7 (captured by the blue line), RK7 (highlighted with the red line), and TL9 (depicted by the black line). (**e**–**g**) Three-dimensional visualizations of the Gibbs free energy landscapes for the complexes formed by PL with RK7, KQ7, and TL9, respectively, offer insights into their thermodynamic stability.

**Figure 3 ijms-26-00756-f003:**
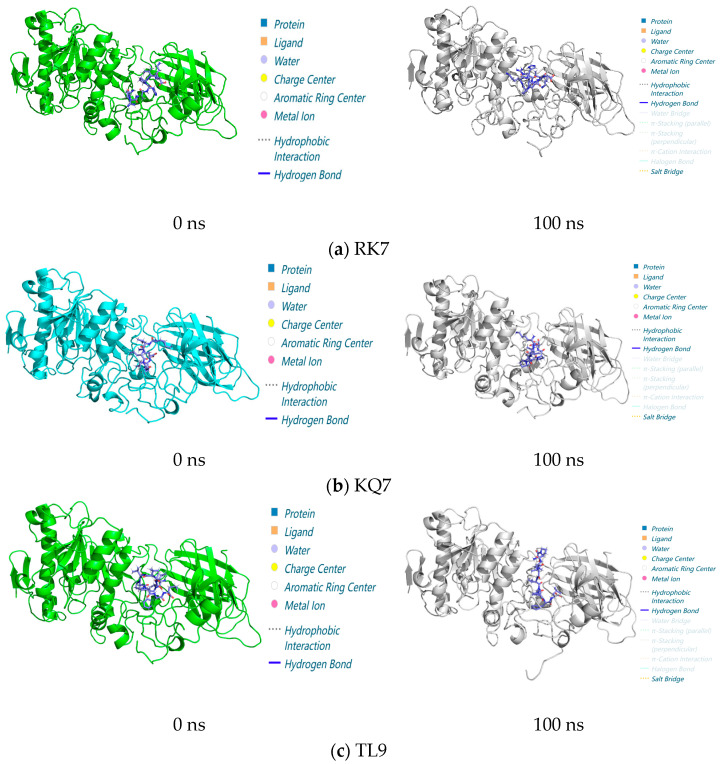
Structure of peptide interaction with PL in the 100 ns molecular dynamics regime: the 3D interaction maps of (**a**) RK7, (**b**) KQ7, and (**c**) TL9 obtained at 0 ns and 100 ns, with elaboration on the interaction diagrams and associated forces between the protein and peptide at 0 ns and 100 ns.

**Figure 4 ijms-26-00756-f004:**
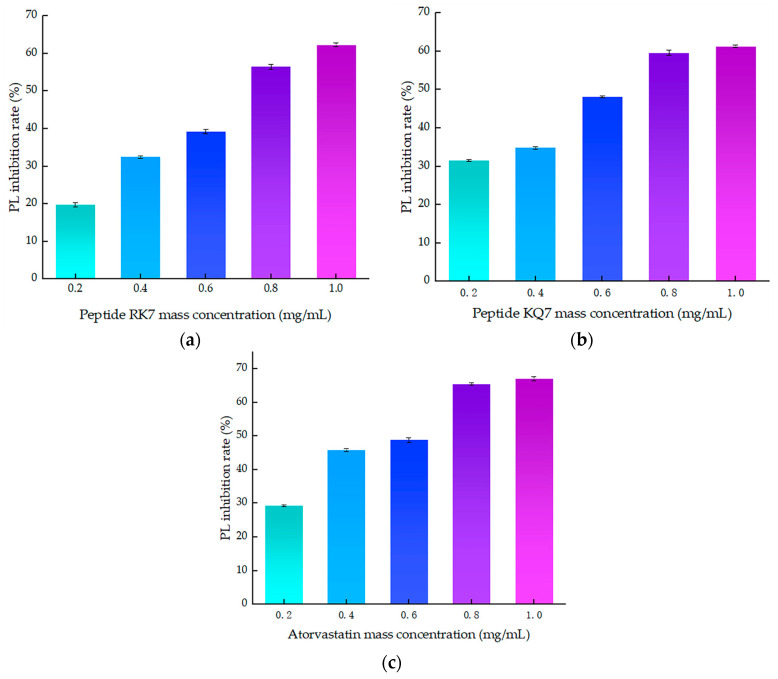
The effect of peptide mass concentration and atorvastatin on PL inhibition rate. (**a**) Effect of RK7 mass concentration on PL inhibition rate. (**b**) Effect of KQ7 mass concentration on PL inhibition rate. (**c**) Effect of atorvastatin mass concentration on PL inhibition rate.

**Figure 5 ijms-26-00756-f005:**
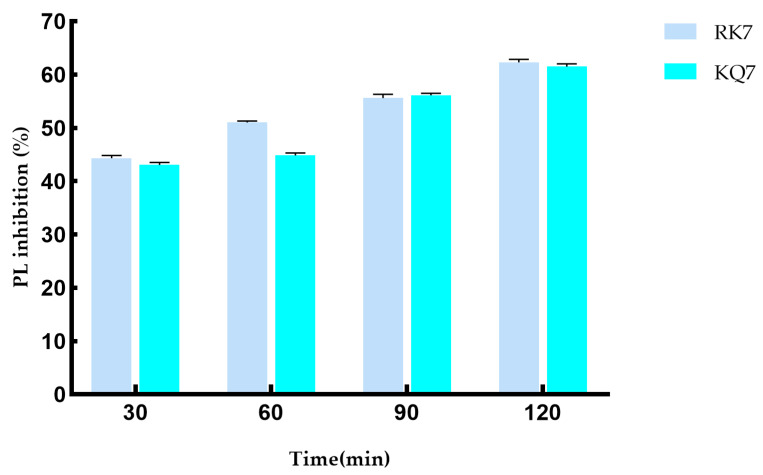
Effect on PL inhibitory activity after simulated gastrointestinal digestion.

**Figure 6 ijms-26-00756-f006:**
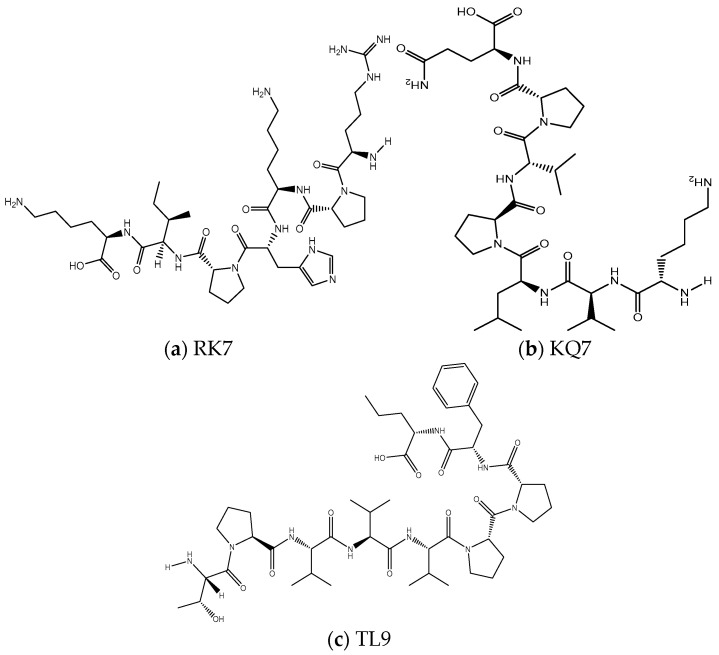
The two-dimensional molecular structures of (**a**) RK7, (**b**) KQ7, and (**c**) TL9.

**Figure 7 ijms-26-00756-f007:**
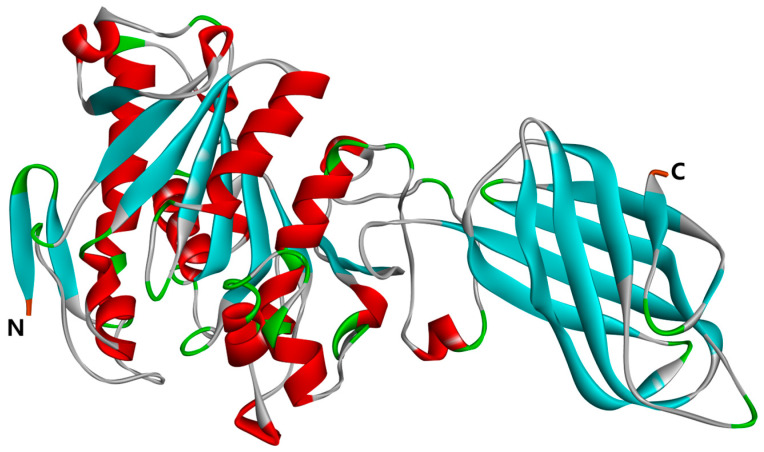
Three-dimensional structure of PL (PDB: 1ETH); the N-terminal and C-terminal positions in the structural domain of PL are labeled in the figure.

**Table 1 ijms-26-00756-t001:** Physicochemical properties of peptides RK7, KQ7, and TL9.

Peptide Sequence	Molecular Weight/(Da)	Isoelectric Point	Net Charge	Hydrophobic Amino Acids	Proportion of Hydrophobic Amino Acids
RK7	874.90	11.57	3	P, I	42.86%
KQ7	779.50	9.84	1	V, L, P	71.42%
TL9	968.19	3.32	0	P, V, F, L	88.91%

**Table 2 ijms-26-00756-t002:** Analysis of the role of interconjugated amino acids: peptides RK7, KQ7, TL9, and PL active site.

	RK7	KQ7	TL9	Atorvastatin
Hydrogen bonds	Val322, Asn320, Gln220, Glu188, Lys198, Gly321, His264	Val325, Asp287, Glu188, Gln220, Ser323, Gly185	Ser323, Gly223, Gln220, Gln22, Glu188, Val222, Pro194, Thr221	Gln220, Glu188, Ser323, Pro194, Gly321, Pro194, Asp193, Val222,
Hydrophobic interactions	Val325	Val322, Phe228, Pro187	Ile211, Ile210, Pro16, Val322, Pro194, Pro187, Leu189	Val322, Arg164, Pro194
Van der Waals forces	Ala197, Gln324, Gly168, Asn167, Ser195, Arg164, Val222, Leu189, Gln220, Pro194, Gly223, Arg191, Asp193, Thr221, Ser323, Val322, Thr221, Tyr327, Lys318, Thr319	Pro194, Asp193, Val222, Ser195, Gln324, Cys286, Tyr327, Asn167, Thr221, Ser219, Leu189, Pro16, Gly185, Thr186, Pro187, Thr186, Arg191, Gln22	Pro209, Ala15, Arg191, Glu188, Asp193, Ser195, Ser219, Gly185, Thr186, Arg23, Gln184, Phe183,	Thr319, Ala197, Lys198, Gly168, Thr196, Asp196, Asn167, Ser195, Asp193, Thr221, Gly223, Glu188, A rg191, Val322,
Electrostatic interactions	Glu188	Glu188	Arg191	Arg191

**Table 3 ijms-26-00756-t003:** MM/GBSA results of PL with RK7, KQ7, and TL9.

(kcal/mol)	RK7	KQ7	TL9
ΔVDWAALS	−71.11 ± 1.65	−63.83 ± 2.46	−49.54 ± 1.49
ΔEEL	−95.60 ± 1.35	−19.66 ± 2.15	−43.28 ± 3.35
ΔEGB	65.11 ± 1.11	91.60 ± 3.06	70.12 ± 1.72
ΔEsurf	−10.12 ± 0.06	−8.39 ± 0.07	−6.88 ± 0.04
ΔGgas	−166.71 ± 2.13	−83.49 ± 3.27	−92.82 ± 3.67
ΔGsolv	111.05 ± 0.52	47.95 ± 0.09	63.23 ± 1.72
ΔTotal	−55.66 ± 2.20	−35.54 ± 3.27	−29.59 ± 4.05

Notes: ΔVDWAALS: van der Waals energy; ΔEEL: electrostatic energy; ΔEGB: polar solvation energy; ΔEsurf: non-polar solvation energy; ΔGgas: energy of the molecular mechanics term (gas-phase energy); ΔGsolv: solvation energy; ΔTotal: combined total energy.

## Data Availability

The original contributions presented in the study are included in the article; further inquiries can be directed to the corresponding author.

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
