# Peer review of "Molecular Docking Studies and In Vitro Activity of Pancreatic Lipase Inhibitors from Yak Milk Cheese"

_ijms, 2025, doi:10.3390/ijms26020756_

Round 1
Reviewer 1 Report (New Reviewer)
Comments and Suggestions for Authors
This manuscript reported to character three oligopeptides from Yak milk cheese through molecular docking and in vitro activity inhibition of pancreatic lipase (LP). The silicon analysis showed these peptides could be bound to LP through multiple interactions, such as hydrogen bond, hydrophobic interaction, salt bridges, etc. In vitro activity assay also showed these peptides were able to inhibit the LP activity. Taken together, this work demonstrated the oligopeptides derived from Yak milk cheese may be of potential as PL inhibitor.
However, the writing of this manuscript is not rigorous and requires significant revisions. Several specific comments were proposed as following:
1. Line 29, HLD at the first occurrence should be spelled fully.
2. Line 34, the authors should cite references to support the statement: “Pancreatic lipase (PL) is responsible for the hydrolysis of 50-70% of total dietary fat in the intestine”
3. Line 63, the conclusion “the inhibitory activity and mechanism of polypeptide on PL were still blank” is not relevant, which is contradiction to the following sentence in Line 64 “Soybean peptides showed lipid-lowering ability by inhibiting PL activity [13]” and also to ref. 21.
4. Line 124-125, revise this sentence.
5. Line 128, what is the likelihood of “target protease binding”.
6. Line 129 also in Table 2 title, what is the “cholesterol lowering peptides”
7. Line 132, what is the “nutrient molecules”?
8. Line 308-317, to delete this section 2.4.1 or transfer it to M&M.
9. Line 319-320, to delete this sentence.
10. Line 323-325, these words are meaningless, can be deleted.
11. Line 329-334, where did the data on the inhibition of “cholesterol esterase” come from?
12. Line 337, how to calculate the IC50 values of RK7 and KQ7. The data present on Fig. 4 is not enough to be used to calculate the IC50 values.
13. Line 347, the authors should cite references to support the conclusion: “Statins, including atorvastatin, exert their inhibitory effects on PL via a multifaceted interaction mechanism”
14. Line 402, “distinguished” should be deleted.
15. Line 409, “leading” should be deleted. Also, the supplier of atorvastatin, Qilu Pharmaceutical is said here in Hainan, China; but in Jinan, China in your previous paper (Wang P et al. Foods 2024, 13, 2970.).
16. Line 464, what is the “Phospholipase (PL)”? Figure 7 could be deleted.
Comments on the Quality of English LanguageN.P.
Author Response
please see attached file

Reviewer 2 Report (New Reviewer)
Comments and Suggestions for Authors
The manuscript seems interesting. A lot of grammar mistakes are present in the manuscript.However, this needs to be revised.

Author Response
Please see the attachment.

Reviewer 3 Report (New Reviewer)
Comments and Suggestions for Authors
the current manuscript describes state of the art methodologies to show that specific peptides isolated from yak cheese can be beneficial in terms of pancreatic lipase inhibition that has been related to serious pathophysiological disorders. Eventhough the revised manuscript seems improved there is still space for improvement.
There are technical errors within the manuscript that must be improved. In addition, in some sections the authors must avoid describing the methods in first person. Other comments are within the attached pdf.
Tables and figures are of good quality.
Based on the overall flow of this work and the findings that were shown I suggest a minor revision.

The English language needs improvement
Round 2
Reviewer 1 Report (New Reviewer)
Comments and Suggestions for Authors
The authors have made proper and satisfactory responses to each point of the comments, and then to improve the manuscript greatly. I have no further comment on this version except one minor suggestion to integrate a same figure/table with its legends in one page.
Reviewer 2 Report (New Reviewer)
Comments and Suggestions for Authors
The manuscript has been revised.
This manuscript is a resubmission of an earlier submission. The following is a list of the peer review reports and author responses from that submission.
Round 1
Reviewer 1 Report
Comments and Suggestions for Authors
It is interesting that the authors investigated the peptides that inhibited PL by virtual screening and in vitro activity assays. But the overall design is quite simple and direct, which is also the limitation of the study.
(1) These analyses of (a) molecular weight, isoelectric point, and hydrophobic amid acids (page 100-126), (b) the binding interactions between pancreatic lipase (PL) and peptides (salt bridges, hydrogen bonds, hydrophobic interactions, van der Waals forces, electrostatic interacting, page 130-239), were too redundant and should be concise.
(2) The authors showed that both peptides showcased a marked upsurge in inhibitory capacity, indicating that the enzymatic degradation by pepsin and trypsin triggers the liberation of potent peptide segments. It is important to identify and isolate these potent peptide segments. If the authors would like to add a section on this, the manuscript would be more complete. That is to say, to elucidate some potent peptide segments actually work after simulated gastrointestinal digestion.
(3) Pancreatic Lipase (PL) catalyzed the degradation of triglycerides on blood carrier lipoproteins into monoacylglycerols and free fatty acids. This was not clearly explained. Have high density lipoproteins or low-density lipoproteins been degraded, and why?
(4) Pay attention to the reference. (a) The journal names of references should be either abbreviated or in full, and should be consistent. Ref 13, 14, 19, and so on. (b) Ref 34, 35, the year is missed.
Comments on the Quality of English LanguageThe English writing skill needs significant improvement, especially in summarizing and generalizing.
Author Response
Response to Reviewer 1 Comments
Dear Reviewer,
We appreciate you for your precious time in reviewing our paper and providing valuable comments. It was your valuable and insightful comments that led to possible improvements in the current version. The authors have carefully considered the comments and tried our best to address every one of them. We hope the manuscript after careful revisions meet your high standards. The authors welcome further constructive comments if any. Below we provide the point-by-point responses to the reviewer’s comments. All modifications in the manuscript are in red (Manuscript ID: ijms-3263676).
Comments 1: (1) These analyses of (a) molecular weight, isoelectric point, and hydrophobic amid acids (page 100-126), (b) the binding interactions between pancreatic lipase (PL) and peptides (salt bridges, hydrogen bonds, hydrophobic interactions, van der Waals forces, electrostatic interacting, page 130-239), were too redundant and should be concise.
Response 1: Thank you very much for raising this question. We think this is a great suggestion. We have made concise modifications to the content and language of sections 2.1. and 2.2.1 in the manuscript based on your suggestions, and highlighted them in red in the manuscript.
Comments 2: The authors showed that both peptides showcased a marked upsurge in inhibitory capacity, indicating that the enzymatic degradation by pepsin and trypsin triggers the liberation of potent peptide segments. It is important to identify and isolate these potent peptide segments. If the authors would like to add a section on this, the manuscript would be more complete. That is to say, to elucidate some potent peptide segments actually work after simulated gastrointestinal digestion.
Response 2: Thank you very much for your constructive suggestion. You pointed out that clarifying the actual role of potent peptides in simulating gastrointestinal digestion would make the manuscript more complete, which is a very accurate and valuable viewpoint. However, this experiment was initiated 2 years ago and the required peptide samples have already been used up. The time required for synthesizing peptides again is approximately 20 working days, so we are unable to add this part of the content in this manuscript at the moment. Referring to Baba et al.'s (2021) study on the inhibitory effect of a new anti - high cholesterol bioactive peptide (BAP) generated from camel whey protein hydrolysate (CWPH) on pancreatic lipase (PL), and Yin et al.'s (2024) experimental design of identifying three bioactive peptides with pancreatic lipase inhibitory activity (LFCMH, RIPAGSPF, YFRPR) in prickly pear seed protein, this study did not design the addition, identification, and isolation of these effective peptides. From the experimental results, although the enzymatic degradation of pepsin and trypsin was not designed to trigger the release, identification, and separation of effective peptides, this study can provide a scientific basis and theoretical guidance for the research of pancreatic lipase inhibitory peptides. But we highly value your suggestion and have made it one of our main directions for future research. We firmly believe that in future research, in - depth exploration of the identification and isolation of these potent peptide segments after simulating gastrointestinal digestion will greatly enrich our understanding of this research field. Thank you again for your careful review and valuable suggestions! In the future, we will rigorously and scientifically design experimental plans to ensure that our research content is richer and more comprehensive.
Comments 3: Pancreatic Lipase (PL) catalyzed the degradation of triglycerides on blood carrier lipoproteins into monoacylglycerols and free fatty acids. This was not clearly explained. Have high density lipoproteins or low-density lipoproteins been degraded, and why?
Response 3: We sincerely appreciate your valuable feedback. After carefully listening to your feedback, we have made revisions and explanations to the introduction section of the manuscript. Pancreatic lipase inhibitors mainly work by reducing the digestion and absorption of fat, and they do not directly target lipoproteins in the blood. Therefore, these inhibitors have relatively limited direct effects on HDL and LDL. However, due to their ability to reduce total fat and triglyceride levels in the blood, they may indirectly affect the metabolism of lipoproteins, such as potentially increasing HDL levels, as less fat may need to be returned to the liver through the reverse cholesterol transport mechanism from surrounding tissues.
Comments 4: Pay attention to the reference. (a) The journal names of references should be either abbreviated or in full, and should be consistent. Ref 13, 14, 19, and so on. (b) Ref 34, 35, the year is missed.
Response 4: Thank you very much for carefully reviewing the manuscript and providing valuable revision suggestions. We have carefully listened to your opinions and made revisions and corrections to the references.
References:
- 1. Baba WN, Mudgil P, Baby B, Vijayan R, Gan CY, Maqsood S. New insights into the cholesterol esterase- and lipase-inhibiting potential of bioactive peptides from camel whey hydrolysates: Identification, characterization, and molecular interaction. J Dairy Sci. 2021;104(7):7393-7405. DOI:10.3168/jds.2020-19868.
- 2. Yin H, Zhu J, Zhong Y, Wang D, Deng Y. Kinetic and thermodynamic-based studies on the interaction mechanism of novel R. roxburghii seed peptides against pancreatic lipase and cholesterol esterase. Food Chem. 2024;447: DOI: 10.1016/j.foodchem.2024.139006.
We appreciate your summary of the manuscript and encouraging comment.
Reviewer 2 Report
Comments and Suggestions for Authors
1. The authors must adequately cite all software and tools used in lines 596-602, such as ProtParam, ensuring they are fully acknowledged with appropriate references.
2. The authors have mentioned using Discovery Studio Professional 20.0 and Discovery Studio Client version 16.1.0. Could you clarify if there is a specific reason for using two different versions or if they are part of the same platform?
3. Please provide detailed docking results, including the LibDock score and binding energies for each compound, to clearly understand the docking interactions and their respective affinities.
4. To further optimize the results, the authors are advised to perform molecular dynamics simulations in triplicate.
5. The overall flow, structure, and length of the paper require improvement. The presentation would benefit from being more concise and focused on the fundamental research objectives, as the current format may hinder clarity and make it difficult for readers to follow.
6. The authors should clarify the differences, similarities, and uniqueness between their recent publication titled “Study on the Inhibitory Effect of Bioactive Peptides Derived from Yak Milk Cheese on Cholesterol Esterase” and the current study. It appears that both works cover similar topics, raising concerns about the novelty of the present research. The authors must demonstrate how this study adds new insights or advances beyond their previous work to establish its distinct contribution to the field.
Author Response
Response to Reviewer 2 Comments
Dear Reviewer,
We appreciate you for your precious time in reviewing our paper and providing valuable comments. It was your valuable and insightful comments that led to possible improvements in the current version. The authors have carefully considered the comments and tried our best to address every one of them. We hope the manuscript after careful revisions meet your high standards. The authors welcome further constructive comments if any. Below we provide the point-by-point responses to the reviewer’s comments. All modifications in the manuscript are in red (Manuscript ID: ijms-3263676).
Comments 1: The authors must adequately cite all software and tools used in lines 596-602, such as ProtParam, ensuring they are fully acknowledged with appropriate references.
Response 1: Thank you very much for raising this question. In section 3.3.1 of the text, we have made an explanation regarding the usage of ExPASy Prot Param and cited reference documents for corroboration.
Comments 2: The authors have mentioned using Discovery Studio Professional 20.0 and Discovery Studio Client version 16.1.0. Could you clarify if there is a specific reason for using two different versions or if they are part of the same platform?
Response 2: Thank you very much for carefully reviewing our manuscript and pointing out any errors. In the article, Discovery Studio Professional 20.0 and Discovery Studio Client version 16.1.0 are the same software. We have made modifications and corrections in this manuscript, and have checked and unified the software names throughout the text.
Comments 3: Please provide detailed docking results, including the LibDock score and binding energies for each compound, to clearly understand the docking interactions and their respective affinities.
Response 3: We are deeply grateful to the reviewers for their meticulous review and valuable comments, which have significantly enhanced the quality of our manuscript. We have added the binding affinity energy data to the polypeptide and PL molecular docking part in the manuscript according to your valuable suggestions, and also provided explanations. It is precisely your valuable suggestions that make our content more comprehensive and richer. Our team would like to thank you again for your valuable suggestions.
Comments 4: To further optimize the results, the authors are advised to perform molecular dynamics simulations in triplicate.
Response 4: We sincerely appreciate your valuable feedback. Your professional insights and constructive feedback have greatly improved the quality of my work. The molecular dynamics data in the manuscript were processed in triplicate, but due to the length of the manuscript, the entire experimental process was not presented. All data presented in the manuscript are the optimal results after the experiment.
Comments 5: The overall flow, structure, and length of the paper require improvement. The presentation would benefit from being more concise and focused on the fundamental research objectives, as the current format may hinder clarity and make it difficult for readers to follow.
Response 5: Thank you very much for your valuable suggestions on our paper. Your feedback is of crucial importance to us in improving our paper. in response to your suggestion that the overall process, structure, and length of the paper need to be improved, we have reorganized the logical order of the article, simplified some redundant discussion parts, and adjusted the layout of the chapters to make the overall process of the paper smoother and the structure clearer and more reasonable. At the same time, we carefully checked the length of the paper and removed some unnecessary content to ensure that each section closely revolves around the core viewpoint. we fully agree with your views on the presentation section. We have made significant revisions to the presentation, removing some distracting elements and focusing on the basic research objectives to improve its clarity and make it easier for readers to understand the content.
Comments 6: The authors should clarify the differences, similarities, and uniqueness between their recent publication titled “Study on the Inhibitory Effect of Bioactive Peptides Derived from Yak Milk Cheese on Cholesterol Esterase” and the current study. It appears that both works cover similar topics, raising concerns about the novelty of the present research. The authors must demonstrate how this study adds new insights or advances beyond their previous work to establish its distinct contribution to the field.
Response 6: We would like to extend our sincere gratitude to the reviewers. you detailed and constructive feedback not only helped us to identify the areas that needed improvement but also provided us with clear directions on how to enhance the overall quality of our study. you professionalism and dedication are truly commendable, and we are extremely thankful for the time and effort you have invested in reviewing our manuscript. Regarding the suggestion you put forward, we make the following reply:
Differences between the two manuscripts of the author:
- 1. Differences between enzymes and peptides in the two manuscripts of this author: In the article published by the author (DOI: 10.3390/foods 13182970), the inhibitory activity of peptides RK7, KQ7, QP13, and VN10 against Cholesterol esterase was tested; in this manuscript, the author uses peptides RK7 and KQ7 to test their inhibitory activity against Pancreatic lipase.
- Peptides RK7 and KQ7 exhibit higher inhibitory activity against pancreatic lipase compared to the previously published article by the author (DOI: 10.3390/foods13182970).
- In the field of cholesterol - lowering activity, relevant research on the inhibitory effect of polypeptides on two different types of target proteases (cholesterol esterase and pancreatic lipase) has not been reported. Therefore, the author uses peptides RK7 and KQ7 to fully study and explore the mechanism of action on the inhibitory activities of these two different enzymes (cholesterol esterase and pancreatic lipase), and will provide relevant theoretical basis and reference for cholesterol - inhibitory activity in the future.
The similarities between the two manuscripts of the author:
The two peptides used in this study (RK7, KQ7) are the same as the peptides RK7 and KQ7 used in the author's article DOI: 10.3390/foods13182970, and the positive control drug used is the same. Therefore, there are similar statements about this part in the writing of this manuscript, and the author has made corresponding modifications in the manuscript.
The uniqueness of the research content in this manuscript lies in:
This study believes that currently, there are multiple enzymes with synergistic effects in the aspect of cholesterol - lowering activity internationally. In our research on cholesterol esterase, we found that polypeptides RK7 and KQ7 showed relatively high inhibitory activities against cholesterol esterase: 22.64% - 61.16%; after the simulated gastrointestinal digestion test, the inhibitory activities against cholesterol esterase were 43.00% - 61.56%. On this basis, we further studied whether polypeptides RK7 and KQ7 also had the same inhibitory effect on pancreatic lipase. Through bioinformatics analysis in the manuscript and verification by in - vitro tests, our experimental research found that polypeptides RK7 and KQ7 also had significant inhibitory effects on pancreatic lipase. The inhibition rate of RK7 was as high as 19.74% - 62.32%, and that of KQ7 was 31.45% - 61.31%; after the simulated gastrointestinal digestion test again, it was found that the inhibitory activities of peptides RK7 and KQ7 on pancreatic lipase were as high as 44.34% - 62.35%. Currently, yak milk cheese - degrading peptides that can significantly inhibit these two enzymes (cholesterol esterase and pancreatic lipase) at the same time have not been reported in the field of cholesterol - lowering activity. Therefore, this study has new findings in this field, and compared with the article already published by the author (DOI: 10.3390/foods13182970), we have made newer progress and contributions to the field of cholesterol - lowering activity.
Thank you very much for your meticulous review. Your suggestions and feedback will prompt us to do more indepth and high-quality work in this research direction.
We appreciate your summary of the manuscript and encouraging comment.
Reviewer 3 Report
Comments and Suggestions for Authors
Dear Authors, The article submitted for review is very interesting and prepared in terms of methodology in a correct manner. The issue itself is also very interesting, but the way of presentation is unacceptable. The article is practically a copy of the content of the previous article by the Authors: 10.3390/foods13182970 The content is almost identical, only (some) symbols are changed. Unfortunately, this is a form of self-plagiarism and as such cannot be accepted for printing.
Author Response
Response to Reviewer 3 Comments
Dear Reviewer,
We appreciate you for your precious time in reviewing our paper and providing valuable comments. It was your valuable and insightful comments that led to possible improvements in the current version. The authors have carefully considered the comments and tried our best to address every one of them. We hope the manuscript after careful revisions meet your high standards. The authors welcome further constructive comments if any. Below we provide the point-by-point responses to the reviewer’s comments. All modifications in the manuscript are in red (Manuscript ID: ijms-3263676).
Comments 1: Dear Authors, The article submitted for review is very interesting and prepared in terms of methodology in a correct manner. The issue itself is also very interesting, but the way of presentation is unacceptable. The article is practically a copy of the content of the previous article by the Authors: 10.3390/foods13182970 The content is almost identical, only (some) symbols are changed. Unfortunately, this is a form of self-plagiarism and as such cannot be accepted for printing.
Response 1: We would like to extend our sincere gratitude to the reviewers. you detailed and constructive feedback not only helped us to identify the areas that needed improvement but also provided us with clear directions on how to enhance the overall quality of our study. you professionalism and dedication are truly commendable, and we are extremely thankful for the time and effort you have invested in reviewing our manuscript. Regarding the suggestion you put forward, we make the following reply:
Differences between the two manuscripts of the author:
- 1. Differences between enzymes and peptides in the two manuscripts of this author: In the article published by the author (DOI: 10.3390/foods 13182970), the inhibitory activity of peptides RK7, KQ7, QP13, and VN10 against Cholesterol esterase was tested; in this manuscript, the author uses peptides RK7 and KQ7 to test their inhibitory activity against Pancreatic lipase.
- Peptides RK7 and KQ7 exhibit higher inhibitory activity against pancreatic lipase compared to the previously published article by the author (DOI: 10.3390/foods13182970).
- In the field of cholesterol - lowering activity, relevant research on the inhibitory effect of polypeptides on two different types of target proteases (cholesterol esterase and pancreatic lipase) has not been reported. Therefore, the author uses peptides RK7 and KQ7 to fully study and explore the mechanism of action on the inhibitory activities of these two different enzymes (cholesterol esterase and pancreatic lipase), and will provide relevant theoretical basis and reference for cholesterol - inhibitory activity in the future.
The similarities between the two manuscripts of the author:
The two peptides used in this study (RK7, KQ7) are the same as the peptides RK7 and KQ7 used in the author's article DOI: 10.3390/foods13182970, and the positive control drug used is the same. Therefore, there are similar statements about this part in the writing of this manuscript, and the author has made corresponding modifications in the manuscript.
The uniqueness of the research content in this manuscript lies in:
This study believes that currently, there are multiple enzymes with synergistic effects in the aspect of cholesterol - lowering activity internationally. In our research on cholesterol esterase, we found that polypeptides RK7 and KQ7 showed relatively high inhibitory activities against cholesterol esterase: 22.64% - 61.16%; after the simulated gastrointestinal digestion test, the inhibitory activities against cholesterol esterase were 43.00% - 61.56%. On this basis, we further studied whether polypeptides RK7 and KQ7 also had the same inhibitory effect on pancreatic lipase. Through bioinformatics analysis in the manuscript and verification by in - vitro tests, our experimental research found that polypeptides RK7 and KQ7 also had significant inhibitory effects on pancreatic lipase. The inhibition rate of RK7 was as high as 19.74% - 62.32%, and that of KQ7 was 31.45% - 61.31%; after the simulated gastrointestinal digestion test again, it was found that the inhibitory activities of peptides RK7 and KQ7 on pancreatic lipase were as high as 44.34% - 62.35%. Currently, yak milk cheese - degrading peptides that can significantly inhibit these two enzymes (cholesterol esterase and pancreatic lipase) at the same time have not been reported in the field of cholesterol - lowering activity. Therefore, this study has new findings in this field, and compared with the article already published by the author (DOI: 10.3390/foods13182970), we have made newer progress and contributions to the field of cholesterol - lowering activity.
We appreciate your summary of the manuscript and encouraging comment.
Round 2
Reviewer 1 Report
Comments and Suggestions for Authors
Looking forward to your further research work.
Author Response
Response to Reviewer 1 Comments
Dear Reviewer,
We appreciate you for your precious time in reviewing our paper and providing valuable comments. It was your valuable and insightful comments that led to possible improvements in the current version. The authors have carefully considered the comments and tried our best to address every one of them. We hope the manuscript after careful revisions meet your high standards. The authors welcome further constructive comments if any. Below we provide the point-by-point responses to the reviewer’s comments. All modifications in the manuscript are in red (Manuscript ID: ijms-3263676).
Comments 1: Looking forward to your further research work.
Response 1: Thank you very much for your interest in our research and your anticipation for further research in the future. We do have some preliminary plans for further research.
Firstly, based on the current research, we plan to add the identification and isolation of these effective peptide segments to your previous manuscript, in order to clarify that the specific effective peptide segments actually play a role in simulating gastrointestinal digestion. We believe this is a very visionary and important opinion, which is of great significance to our current research topic. Therefore, we will incorporate this part into the future research on the biological activity of related peptides. This will help us gain a deeper understanding of the mechanism of action after peptide digestion.
Secondly, we plan to introduce new molecular docking and kinetic analysis methods. At present, our research mainly focuses on the analysis and experimentation of peptide biological activity inhibition target enzymes. However, with the continuous development of the research field, we believe that there are updated and accurate software to analyze docking and kinetic data. Therefore, we plan to include it in future research. This method has unique advantages in future bioinformatics and is expected to bring new perspectives and deeper insights to our research.
Finally, we also hope to engage in interdisciplinary research collaboration. We recognize the interdisciplinary nature of our research topic in pharmacy, and by collaborating with experts in this field, we can integrate theories and methods from different disciplines, opening up new avenues for our research.
Thank you again for your expectations and suggestions. We believe that through these further research efforts, we can make more meaningful contributions to the development of this field.
Wishing you all the best!
We appreciate your summary of the manuscript and encouraging comment.
Reviewer 2 Report
Comments and Suggestions for Authors
Authors have revised manuscript satisfactorily
Author Response
Response to Reviewer 2 Comments
Dear Reviewer,
We appreciate you for your precious time in reviewing our paper and providing valuable comments. It was your valuable and insightful comments that led to possible improvements in the current version. The authors have carefully considered the comments and tried our best to address every one of them. We hope the manuscript after careful revisions meet your high standards. The authors welcome further constructive comments if any. Below we provide the point-by-point responses to the reviewer’s comments. All modifications in the manuscript are in red (Manuscript ID: ijms-3263676).
Comments 1: Authors have revised manuscript satisfactorily.
Response 1:Thank you very much for recognising our revision of the manuscript. This result could not be achieved without your previous valuable suggestions, which played a crucial role in improving the quality of the manuscript.
We have carefully considered each of your suggestions during the revision process and have thoroughly checked and improved the manuscript. In this process, we also gained a deeper understanding of the relevant content, which will help us do better in our subsequent research and writing.
Thank you again for your careful review and positive feedback, and we look forward to the subsequent progress of the manuscript.
We wish you all the best in your work!
We appreciate your summary of the manuscript and encouraging comment.
Reviewer 3 Report
Comments and Suggestions for Authors
The Authors provide improved version. The manuscript looks much better. Could the authors explain:
1. The molecular dynamics was performed in one repetion to each ligand? Or was it repeated?
2. Fig. 3 - could you provide version with better resolution? It is hard to analyze. It can be moved to supplementary materials.
Author Response
Response to Reviewer 3 Comments
Dear Reviewer,
We sincerely thank the editor and all reviewers for their valuable feedback that we have used to improve the quality of our manuscript.On behalf of all the contributing authors, I would like to express our sincere appreciations of your letter constructive comments concerning our article (Manuscript ijms-3263676). These comments are all valuable and helpful for improving our article. Based on your comments, we have revised the manuscript accordingly. Point-by-point responses to the nice reviewers are listed below this letter.
Comments 1: The molecular dynamics was performed in one repetion to each ligand? Or was it repeated?
Response 1: Thank you again sincerely for reviewing the manuscript. We have the following response to the question you raised. In this manuscript, molecular dynamics studies in which a complex formed by a protein with a small molecule peptide is studied as a separate kinetic system for 0-100 ns.
Comments 2: Fig. 3 - could you provide version with better resolution? It is hard to analyze. It can be moved to supplementary materials.
Response 2: Thank you very much for your suggestions about the clarity of the images in the manuscript. We have improved the clarity of all the images in Figure 3 in the manuscript to ensure that you can read the manuscript accurately.
We appreciate your summary of the manuscript and encouraging comment.
Round 3
Reviewer 3 Report
Comments and Suggestions for Authors
The whole conclusion part is still a copy of an earlier publication of Authors. Only symbols of ligands are changed. It is a plagiarsim and must be changed.
The Authors have not respond to my last question about repetition of molecular dynamics. Was it performed once for each ligand or in more repetition?
Round 4
Reviewer 3 Report
Comments and Suggestions for Authors
The article has been improved, it can be accepted for publications.